# Pianist Transformer: Towards Expressive Piano Performance Rendering via Scalable Self-Supervised Pre-Training

Hong-Jie You [1 2]   Jie-Jing Shao [1]   Xiao-Wen Yang [1 2]   Lin-Han Jia [1]   Lan-Zhe Guo [1 3]   Yu-Feng Li [1 2]

## Abstract

Existing methods for expressive music performance rendering, a conditional generation task that aims to generate a human-like performance from a symbolic score, rely on supervised learning over small labeled datasets, which limits scaling of both data volume and model size, despite the availability of vast unlabeled music, as in vision and language. To address this gap, we introduce Pianist Transformer, with three key contributions: 1) introducing large-scale self-supervised learning into expressive piano performance rendering through a unified Musical Instrument Digital Interface (MIDI) representation, enabling pre-training on 10B tokens of unlabeled MIDI data; 2) an efficient asymmetric Transformer with note-level compression, substantially improving training efficiency, memory usage, and inference speed for long-context music modeling; 3) a state-of-the-art rendering model with an editable workflow, achieving strong objective and subjective results and enabling integration into real-world music production workflows. Overall, Pianist Transformer outlines a scalable path toward human-like performance synthesis in the music domain. Code, audio samples, and model checkpoints are available on our project page: https://yhj137.github.io/pianist-transformer-demo/.

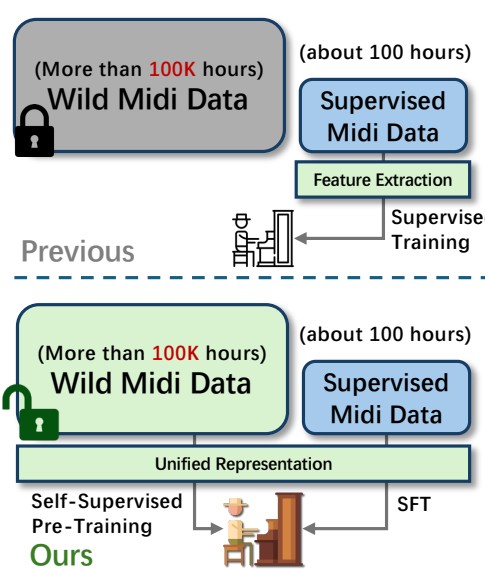

*Figure 1.* **From Supervised to Self-Supervised Learning for Expressive Piano Performance Rendering. (Top) Previous Supervised Paradigm:** Existing systems operate under a strictly supervised pipeline that depends on scarce aligned datasets ($\approx$ 100 hours) and cannot exploit the vast in-the-wild MIDI corpus ($> 100K$ hours). This reliance on explicit structural features fundamentally limits scalability. **(Bottom) Our Scalable Self-Supervised Paradigm:** Pianist Transformer enables large-scale self-supervised pre-training on over 100K hours of unaligned MIDI to acquire musical priors, followed by supervised fine-tuning for expressive piano performance rendering.

## 1. Introduction

In the music domain, expressive performance rendering aims to automatically generate a human-like musical performance from a symbolic score. This task goes beyond mere pitch-and-rhythm accuracy to capture the subtle variations in timing, dynamics, articulation, and pedaling that shape musical expression. The core challenge lies in computationally modeling the intricate mapping from a score's underlying musical structures, such as its melody and harmony, to these expressive choices. For decades, research from probabilistic models (Teramura et al., 2008) to modern deep learning approaches using RNNs (Jeong et al., 2019b) and Transformers (Borovik & Viro, 2023) has predominantly relied on a supervised paradigm. This paradigm, however, faces a persistent bottleneck: the aligned score-performance supervised datasets are typically labor-intensive and expensive to

[1]State Key Laboratory of Novel Software Technology, Nanjing University, Nanjing 210023, China [2]School of Artificial Intelligence, Nanjing University, Nanjing 210023, China [3]School of Intelligence Science and Technology, Nanjing University, Suzhou 215163, China. Correspondence to: Yu-Feng Li <liyf@nju.edu.cn>.

*Proceedings of the $43^{rd}$ International Conference on Machine Learning*, Seoul, South Korea. PMLR 306, 2026. Copyright 2026 by the author(s).

scale.

To maximize the utilization of the limited dataset, existing works often adopt asymmetric, specialized representations, injecting rich structural descriptors on the score side (e.g., measures, meter) (Jeong et al., 2019b; Maezawa et al., 2019; Borovik & Viro, 2023). This improves label efficiency because each labeled example delivers explicit structural cues rather than forcing the model to infer them. However, these descriptors require a notated score and a score–performance alignment. In contrast, performance MIDI is typically captured from digital-piano recordings or produced by AI transcription and thus consists of a stream of note events without explicit measures, meter, or a usable tempo map. As a result, such methods cannot compute their required structural features for the vast, unaligned corpora of performance-only MIDI, making them ill-suited for leveraging unsupervised data at scale (Figure 1, top). Renault et al. (2023) explores an adversarial, cycle-consistent architecture that disentangles score content from performance style, and a score-to-audio generator learns to render expressive piano audio from unaligned data against a realism discriminator. Nevertheless, adversarial training is challenging to scale due to complex training dynamics, and the resulting generation quality has so far been limited. A more stable and scalable paradigm is desirable to truly harness the potential of the vast, unaligned corpora of in-the-wild MIDI performances that remain largely untapped (Figure 1, bottom).

In this paper, we present Pianist Transformer, a model for expressive performance rendering trained with large-scale unlabeled MIDI corpus. Our main contributions are as follows:

**Paradigm Shift in Piano Performance Rendering.** We introduce large-scale self-supervised learning into expressive piano performance rendering, shifting the task from data-limited supervised modeling to scalable pre-training on unlabeled data. By formulating musical scores and performances under a unified MIDI representation, our approach avoids explicit modeling of score-specific structure and treats score and performance MIDI uniformly, enabling self-supervised training on 10B tokens of in-the-wild MIDI data. As a result, our approach alleviates the data bottleneck and demonstrates the feasibility of large-scale self-supervised pre-training in expressive piano performance rendering.

**Efficient Architecture with Synergistic Design.** To address the inherent long-context and efficiency challenges of music modeling, we design an efficient architecture combining an asymmetric encoder–decoder Transformer with note-level sequence compression. While each component independently improves efficiency, their combination exhibits a non-additive effect, yielding gains in training speed and memory reduction that exceed the product of their individual improvements. In particular, the joint design achieves up to

3.13× training speedup and reduces training VRAM usage to 0.38× of a symmetric uncompressed baseline. In addition, the asymmetric architecture alone enables up to 2.1× faster inference, making large-scale training and low-latency rendering practically feasible.

**SOTA Rendering Model and Editable Workflow.** We introduce a state-of-the-art model for expressive piano performance rendering, establishing new performance levels compared to prior approaches. This is supported by consistent improvements across objective evaluation metrics and our subjective listening study, where listeners show no statistically significant ability to distinguish our generated performances from human recordings. We further introduce Expressive Tempo Mapping, a post-processing algorithm mapping expressive timing to an editable tempo curve. This produces an editable format suitable for real-world use while preserving expressive timing, serving as a bridge between model outputs and practical music production.

## 2. Related Work

**Piano Performance Rendering.** The goal of performance rendering is to synthesize an expressive, human-like performance from a symbolic score. The field has evolved from early rule-based (Sundberg et al., 1983) and statistical models (Teramura et al., 2008; Flossmann et al., 2013; Kim et al., 2013) to deep learning architectures based on RNNs (Cancino-Chacón & Grachten, 2016), variational autoencoders (Maezawa et al., 2019), graph neural networks (Jeong et al., 2019b), and Transformers. For instance, recent work such as ScorePerformer (Borovik & Viro, 2023) has focused on fine-grained stylistic control, a goal complementary to our focus on scalable pre-training. Despite these architectural innovations, progress has been bottlenecked by the supervised learning paradigm; the small, costly aligned datasets it requires are insufficient for models to learn the complex mapping from musical structure to expressive nuance. This reliance limits model scalability and generalization. While recent work has explored adversarial training on unpaired data to bypass alignment (Renault et al., 2023), challenges with training stability and quality remain. This underscores the need for a robust paradigm that can effectively leverage vast, unaligned data, which we propose through large-scale self-supervised pre-training.

**Self-Supervised Learning in Music.** Self-supervised pre-training, a dominant paradigm in NLP (Devlin et al., 2019; Brown et al., 2020) and computer vision (Chen et al., 2020; He et al., 2022), has also been adapted for music. In the symbolic domain, early efforts like MusicBERT (Zeng et al., 2021) applied masked language modeling to MIDI for understanding tasks. This approach has recently been scaled up significantly: Bradshaw et al. (2025) pre-trained on large

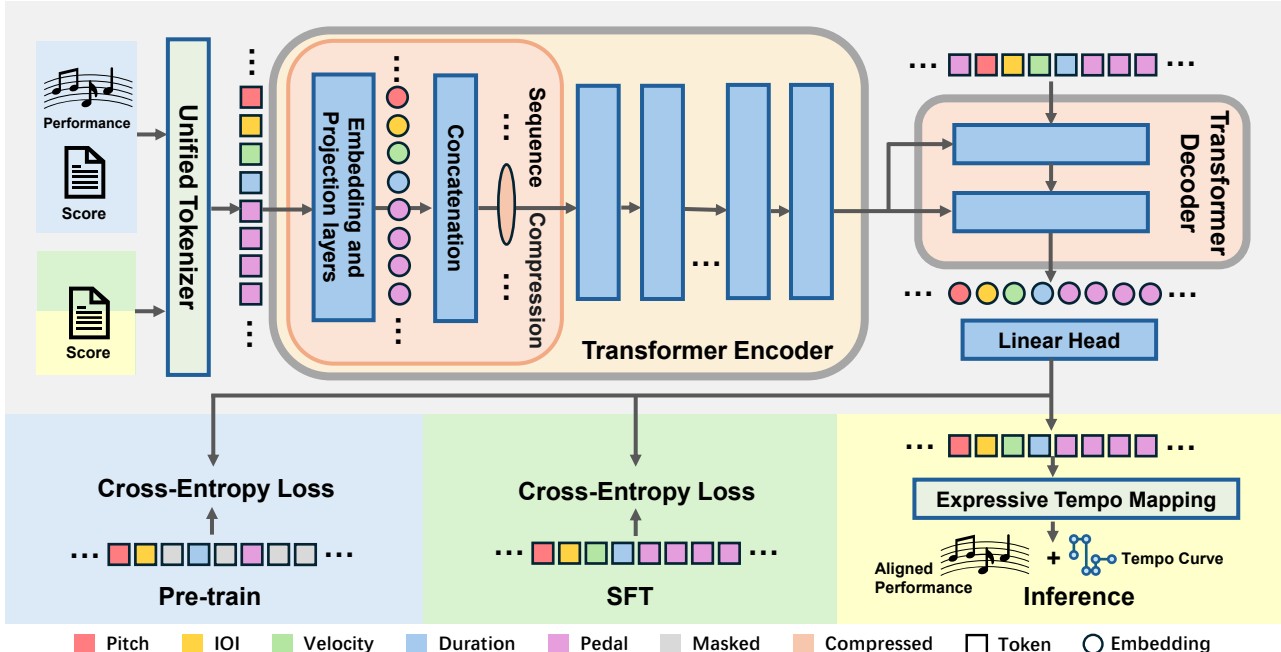

*Figure 2.* **The overall architecture and workflow of Pianist Transformer.** Our framework processes all MIDI data through a **Unified Tokenizer**, enabling a two-stage training process. The core model is an **asymmetric Transformer** with **Encoder Sequence Compression** for efficient processing of long musical scores. The workflow consists of three stages: **(1) Pre-train**: The model learns foundational musical context from a massive unlabeled corpus via a masked denoising objective, where it takes a masked token sequence as input and predicts the original sequence. **(2) SFT**: Supervised Fine-Tuning adapts the model to map musical context to expressive nuances using aligned score-performance pairs, where it takes the score tokens as input and predicts the corresponding performance tokens. **(3) Inference**: The model takes a score input and then generates a performance, which is then made editable for DAWs by our **Expressive Tempo Mapping** algorithm.

piano corpora for tasks like melody continuation, while foundation models like Moonbeam (Guo & Dixon, 2025) have been trained on billions of tokens for diverse conditional generation. Parallel efforts also exist for raw audio using contrastive or reconstruction objectives (Spijkervet & Burgoyne, 2021; Hawthorne et al., 2022). However, the application of self-supervised pre-training to the specific task of expressive performance rendering is largely unexplored. While existing self-supervised models excel at learning high-level musical semantics for tasks like generation or classification, performance rendering is a distinct, fine-grained challenge centered on modeling subtle expressive details. Whether the benefits of large-scale self-supervised pre-training can successfully transfer to this nuanced, performance-level domain is an open question that motivates our work.

# 3. The Pianist Transformer Framework

Our goal is to develop a powerful piano performance rendering system that can leverage large-scale, unlabeled data through a self-supervised pre-training paradigm. This section details our approach, beginning with the core of our methodology: a unified data representation that enables large-scale pre-training. We then describe the Transformer-

based architecture and the two-stage training strategy built upon this representation. Finally, we introduce a novel post-processing algorithm that ensures the model's output is editable and practical for musicians.

## 3.1. Unified MIDI Representation

A fundamental challenge in applying self-supervised learning to performance rendering is the disparity between structured score data and expressive performance data. Specifically, scores represent music with metrical timing (e.g., quarter notes, eighth notes) and categorical dynamics (e.g., p, mf, f), while performances are captured as streams of events with absolute timing in milliseconds and continuous velocity values. To overcome this, we propose a unified, event-based token representation that treats both formats identically, enabling them to be mixed in a single, massive pre-training corpus.

We represent each musical note as a sequence of eight tokens. This sequence captures the note's Pitch, Velocity, Duration, and the Inter-Onset Interval (IOI) from the previous note, with timing information quantized at millisecond resolution. To model nuanced pedal control, we include four additional Pedal tokens, which represent the sustain

pedal state at uniformly sampled points within the note's IOI window. Full details are provided in Appendix A.2.2.

Crucially, this representation, avoiding reliance on high-level musical concepts like measures or beats, unlocks large-scale pre-training on unaligned MIDI and empowers the model to uncover musical principles, from melodic contours to harmonic progressions, through statistical regularities.

### 3.2. Architecture for Efficient Long-Sequence Modeling

We employ an Encoder-Decoder Transformer, but its standard $O(N^2)$ self-attention complexity presents a critical bottleneck for long musical sequences, which often exceed thousands of tokens. To enable efficient training and inference, we introduce two synergistic architectural modifications: Encoder Sequence Compression and an Asymmetric Layer Allocation.

**Encoder Sequence Compression.** Leveraging the fixed 8-token structure of each note, we compress the encoder's input sequence. We first project and then aggregate the eight embeddings of a single note into one consolidated vector. This note-level aggregation reduces the sequence length by a factor of 8, resulting in a substantial reduction in self-attention computation: the attention matrix size is reduced from $N^2$ to $(N/8)^2$, i.e., by a factor of 64. As a result, the encoder can efficiently process much longer sequences, capturing the global context essential for rendering.

**Asymmetric Encoder-Decoder Architecture.** We employ a deliberately asymmetric architecture with a deep 10-layer encoder and a lightweight 2-layer decoder to maximize efficiency. Combined with sequence compression, this design concentrates most computation in a single, highly parallelizable encoding pass, substantially improving training speed and reducing memory usage for both training and inference. Importantly, despite its shallow depth, the decoder is sufficient to generate high-quality expressive performances when conditioned on the strong contextual representations produced by the encoder. We further analyze the impact of this architectural choice in Section 4.6. A detailed analysis of the synergistic acceleration effects between architectural asymmetry and sequence compression is provided in Appendix E.

### 3.3. Two-Stage Training for Expressive Rendering

Our training paradigm directly addresses the core challenge of expressive rendering: modeling the complex dependency between a score's musical structure and the nuances of human performance. To achieve this, our training proceeds in two stages: first, learning to comprehend musical context, and second, learning to translate that context into an expressive performance.

**Self-Supervised Pre-training.** The initial pre-training stage builds an understanding of the implicit context guiding human expression. We employ a self-supervised masked denoising objective on our massive, unlabeled MIDI corpus. By learning to reconstruct the original music pieces from their corrupted context, the model is compelled to internalize the deep structural cues such as harmonic function and melodic direction that inform performance choices.

The objective is to minimize the negative log-likelihood of the original tokens at the masked positions:

$$\mathcal{L}_{\text{pre-train}} = -\sum_{i \in M} \log p(x_i | X_{corr}, X_{<i})$$

where $M$ is the set of indices of the masked tokens, and $p(x_i | X_{corr}, X_{<i})$ is the probability of predicting the original token $x_i$ given the corrupted input $X_{corr}$ and the ground-truth prefix $X_{<i}$.

**Supervised Fine-tuning.** With a model that comprehends musical context, we then perform Supervised Fine-Tuning (SFT) to teach it how to translate this understanding into a performance. This stage learns to map the latent representations of the score to continuous performance parameters that govern expressive timing, dynamics, and articulation.

The SFT is framed as a sequence-to-sequence learning task on aligned score-performance pairs. The encoder processes the score's token sequence, while the decoder is trained to autoregressively generate the corresponding performance sequence by minimizing a standard cross-entropy loss.

### 3.4. Post-processing: Expressive Tempo Mapping

A key challenge for practical application is that raw model outputs, with timings in absolute milliseconds, lack compatibility with standard music software. These performances do not align with the metrical grid of a Digital Audio Workstation (DAW), hindering editability. To bridge this gap between AI generation and modern music production workflows, we introduce a novel post-processing algorithm, Expressive Tempo Mapping.

This algorithm, detailed in Appendix B, translates the performance's expressive timing deviations into a dynamic tempo curve. It then realigns all note and pedal events to a musical grid governed by this new tempo curve. The process preserves the expressive nuances of the generated performance while restoring the structural alignment essential for editing. The final output is a MIDI file that is both musically expressive and fully editable in any standard DAW.

## 4. Experiments

We conduct a comprehensive set of experiments to evaluate our proposed Pianist Transformer. Our evaluation is guided

*Table 1.* **Objective evaluation results on the ASAP test set.** We compare our **Pianist Transformer** against baselines using JS Divergence and Intersection Area. For JS Div, lower is better (↓). For Intersection, higher is better (↑). The "Overall" columns report the average scores across the four expressive dimensions. Our model achieves the best performance on most metrics among the models, outperforming prior SOTA and demonstrating the profound impact of pre-training.

| Model | Velocity | | Duration | | IOI | | Pedal | | Overall (Avg.) | |
|---|---|---|---|---|---|---|---|---|---|---|
| | JS Div (↓) | Inter. (↑) | JS Div (↓) | Inter. (↑) | JS Div (↓) | Inter. (↑) | JS Div (↓) | Inter. (↑) | JS Div (↓) | Inter. (↑) |
| Human | 0.0427 | 0.9724 | 0.0438 | 0.9655 | 0.0535 | 0.9496 | 0.0244 | 0.9771 | 0.0411 | 0.9662 |
| Score | 0.7492 | 0.2255 | 0.6868 | 0.3152 | 0.7706 | 0.2106 | 0.4281 | 0.5467 | 0.6587 | 0.3245 |
| ScorePerformer | 0.4004 | 0.6256 | 0.3116 | 0.7126 | 0.4555 | 0.6071 | 0.6174 | 0.4164 | 0.4462 | 0.5904 |
| VirtuosoNet-ISGN | 0.2574 | 0.7981 | 0.2321 | 0.7903 | 0.5441 | 0.4928 | **0.0829** | **0.9410** | 0.2791 | 0.7556 |
| VirtuosoNet-Han | 0.2407 | 0.8132 | 0.3438 | 0.6744 | 0.4170 | 0.6374 | 0.1339 | 0.8507 | 0.2839 | 0.7439 |
| Pianist Transformer (w/o PT) | 0.5363 | 0.4826 | 0.5399 | 0.4886 | 0.2789 | 0.7360 | 0.2860 | 0.7054 | 0.4103 | 0.6032 |
| **Pianist Transformer (Ours)** | **0.1805** | **0.8517** | **0.1879** | **0.8303** | **0.1740** | **0.8292** | 0.1111 | 0.8893 | **0.1634** | **0.8501** |

by three central questions. **First**, how does Pianist Transformer perform against existing methods when judged by both objective metrics and subjective human evaluation? **Second**, to what extent does large-scale self-supervised pre-training contribute to the final performance of a rendering model? **And third**, what architectural design choices are critical to performance, and how does the model scale with respect to model and data size? The following sections are structured to address each of these questions in turn.

### 4.1. Experimental Setup

We pre-train our model on a massive 10-billion-token corpus aggregated from several public MIDI datasets. For supervised fine-tuning and evaluation, we use the ASAP dataset (Foscarin et al., 2020) with a strict piece-wise split. Our Pianist Transformer is compared against strong baselines, including VirtuosoNet-HAN (Jeong et al., 2019a), VirtuosoNet-ISGN (Jeong et al., 2019b) and ScorePerformer (Borovik & Viro, 2023), as well as the unexpressive Score MIDI and ground-truth Human performances.

We evaluate all models using a suite of objective and subjective measures. Objectively, we assess distributional similarity to human performances using Jensen-Shannon (JS) Divergence and Intersection Area across four key expressive dimensions: Velocity, Duration, IOI and Pedal. Subjectively, we conduct a comprehensive listening study that evaluates overall preference as well as multiple expressive dimensions, including human-likeness, dynamics, rhythm, and articulation. Further details on the datasets, baseline implementations, and evaluation protocols are provided in Appendix A and Appendix C.

### 4.2. Pianist Transformer Achieves SOTA Results

We compare Pianist Transformer against prior state-of-the-art models. As shown in Table 1, our model consistently outperforms existing methods across both objective metrics and overall aggregated scores.

Among all generative models, Pianist Transformer achieves the best results on 6 out of 8 evaluation metrics and on both overall averages. Notably, it substantially narrows the gap to the Human ground truth. For example, its overall JS Divergence of 0.1634 represents a large improvement over the strongest baseline, VirtuosoNet-ISGN (0.2791), indicating that the expressive distributions generated by our model are significantly closer to those of human performances.

Examining the per-dimension results, the largest gains are observed in Duration and IOI, which play a central role in shaping musical timing and rhythmic feel. Pianist Transformer achieves markedly lower JS Divergence scores on these dimensions than all baseline methods, highlighting its strength in modeling fine-grained temporal expression.

It is worth noting that VirtuosoNet-ISGN achieves a better score on the Pedal metrics, likely due to its specialized architectural design. Nevertheless, Pianist Transformer still produces high-quality pedaling, achieving a competitive JS Divergence of 0.1111 and outperforming the remaining baselines. Overall, these results demonstrate that our approach delivers strong and well-balanced performance across expressive dimensions.

### 4.3. Subjective Evaluations Demonstrate Human-Level Performance

While objective metrics quantify statistical similarity, they do not fully capture perceptual aspects of musical expression. We therefore conduct a subjective listening study to evaluate the model's performance across multiple expressive dimensions, as well as its degree of human-likeness and overall preference by human listeners.

**Study Design.** We conducted a subjective listening study with careful controls to ensure reliable human evaluation. We recruited 57 participants with diverse musical backgrounds and retained 39 valid responses after filtering based on attention checks and completion time. Participants rated and ranked five anonymized performance versions (our model, two baselines, Score, and Human) across six 15-

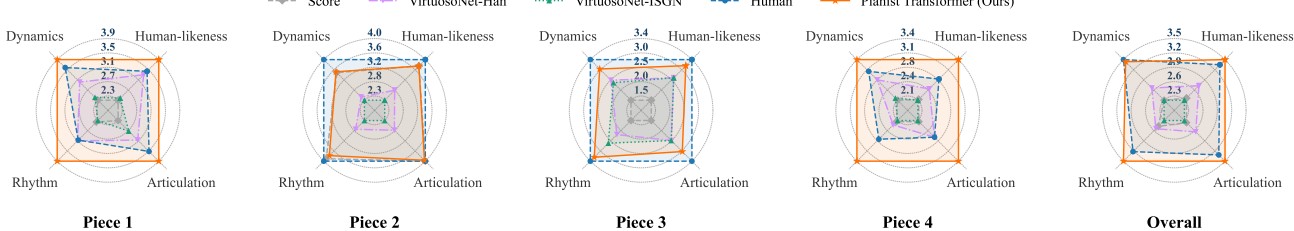

*Figure 3.* **Multi-dimensional Subjective Ratings (Normalized).** A radar chart visualizing the average scores on a 5-point scale for four expressive dimensions. **Pianist Transformer** exhibits a profile that closely mirrors the **Human** performance, indicating a well-balanced and high-quality rendering across all aspects. The area covered by ours is substantially larger than all of other baselines.

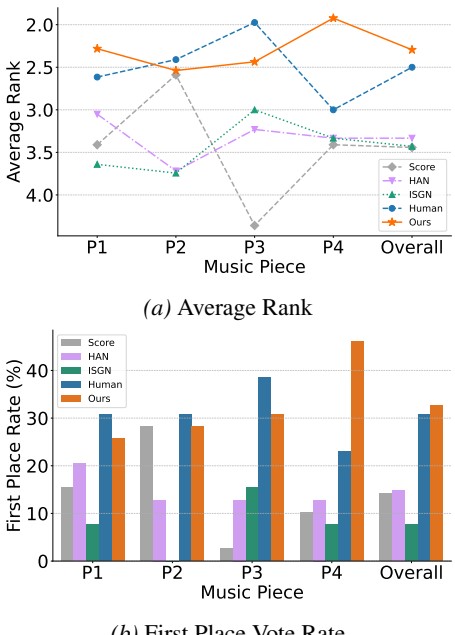

*(a)* Average Rank

*(b)* First Place Vote Rate

*Figure 4.* **Subjective Preference Ranking Results.** The evaluation includes pieces by Haydn (P1), Beethoven (P2), Chopin (P3), and Bach (P4). (a) The average rank of **Pianist Transformer** is comparable to that of the Human performance and significantly better than all baseline models. (b) **Pianist Transformer** achieves the highest first-place vote rate among all generative systems.

second music clips spanning styles from Baroque to modern Pop. To reduce potential bias, the presentation order of all performances was randomized for each participant. Further details of the study design, including participant demographics, clip selection, and reliability analyses, are provided in Appendix C.

**Main Results.** The listening study results, summarized in Figure 4, reveal a clear and consistent listener preference for Pianist Transformer. As listener preference provides the most direct assessment of perceptual quality in blind evaluations, our model was consistently ranked as the top-performing system among all generative methods.

As shown in Figure 4b, Pianist Transformer achieves the highest first-place vote rate (32.7%), substantially outperforming all baseline models and slightly exceeding the human reference recordings (30.8%). This indicates that, under blind listening conditions, the model's renderings are frequently preferred by listeners.

This observation is further supported by the average ranking results in Figure 4a. Pianist Transformer attains the best overall average rank (2.29), outperforming all baseline systems. Statistical analysis using two-sided paired t-tests confirms that the model is rated significantly higher than VirtuosoNet-ISGN ($p < 0.001$), VirtuosoNet-Han ($p < 0.001$), and the Score baseline ($p < 0.001$). While the difference between our model and the human performance is not statistically significant ($p = 0.21$), these results demonstrate that the model achieves human-comparable quality while frequently being preferred by listeners in our study.

**Multi-dimensional Quality.** To better understand the source of this strong listener preference, we analyze the multi-dimensional perceptual ratings across expressive attributes.

As shown in the radar chart (Figure 3), Pianist Transformer exhibits a consistently strong expressive profile across all perceptual dimensions. Notably, with the exception of Dynamics, the model receives higher average ratings than the human reference in Rhythm (3.44 vs. 3.21), Articulation (3.38 vs. 3.24), and Human-likeness (3.43 vs. 3.29), while substantially outperforming all baseline systems across all dimensions.

This pattern indicates that the model delivers a more uniformly high level of perceptual quality across expressive attributes, whereas individual human performances exhibit greater variability across dimensions. Under subjective evaluation, such cross-dimensional consistency appears to align well with listener expectations, contributing to the strong preference observed for Pianist Transformer.

**Stylistic Robustness.** Beyond overall preference, we further examine the robustness of the model across musical styles and historical periods. As shown in Figure 5, baseline methods exhibit strong style dependency, with notice-

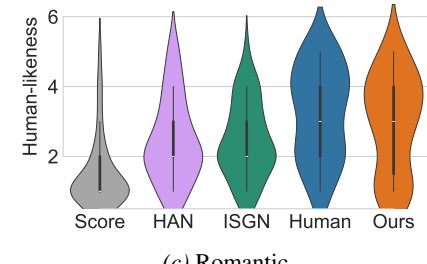

*Figure 5.* **Analysis of Human-likeness Scores Across Musical Styles.** Violin plots show the distribution of Human-likeness ratings for each model, grouped by historical period. (a, b) For Baroque and Classical music, the performance of baseline models degrades significantly, sometimes falling below the Score baseline. (c) While baselines perform better on Romantic music, **Pianist Transformer** maintains a consistently high level of performance across all styles.

able performance degradation on Baroque and Classical pieces. In contrast, Pianist Transformer maintains high human-likeness ratings across the evaluated styles.

While the number of clips in each style group is limited, these results suggest a potential benefit of large-scale pre-training on diverse musical data in improving the model's adaptability to different musical styles. Additional evidence of this generalization ability is provided by an out-of-domain case study on pop music in Appendix C.3, where the preference advantage of Pianist Transformer becomes even more pronounced.

### 4.4. Pre-training Substantially Improves Performance

To quantify the impact of large-scale self-supervised pre-training, we perform a controlled ablation comparing our full Pianist Transformer with an identical model trained by the same supervised objective but without pre-training (w/o PT). This setup isolates the effect of pre-training and reveals its crucial role in expressive performance modeling.

**Quantitative Impact of Pre-training.** The results demonstrate a clear and consistent advantage of pre-training across all objective metrics. As reported in Table 1, the overall Intersection Area improves from 0.6032 to 0.8501, corresponding to a 40.9% relative gain. At the same time, JS Divergence is substantially reduced for velocity, duration, IOI, and pedal, with reductions ranging from 37.6% to over 65%. These improvements hold uniformly across all expressive dimensions, indicating a markedly closer match between the generated performances and human expressive distributions.

**Representation Analysis of Pre-training Effects.** To better understand the source of the performance gap observed in Table 1, we analyze the structure of the encoder representation space using t-SNE visualizations, where samples are colored according to their historical musical periods. Figure 6 compares representations learned with supervised training

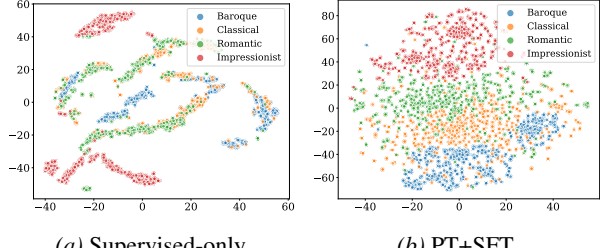

*(a)* Supervised-only      *(b)* PT+SFT

*Figure 6.* **t-SNE Visualization of Encoder Latent Representations across Musical Styles.** We visualize note-level encoder embeddings across musical periods. (a) Without self-supervised pre-training, representations exhibit fragmented and entangled structures with limited separation across periods. (b) With large-scale self-supervised pre-training, the representation space becomes smoother and more structured, with musical periods forming an ordered and continuous arrangement.

only (Supervised-only) and with large-scale self-supervised pre-training followed by fine-tuning (PT+SFT).

In the Supervised-only setting, the representation space exhibits a highly fragmented geometry, characterized by scattered local clusters with limited global continuity. Semantically related performances are distributed across isolated regions, suggesting that expressive attributes are encoded in a brittle and locally constrained manner. Such fragmentation implies that the decoder must learn to map from irregular and discontinuous manifolds to coherent expressive outputs, increasing sensitivity to small perturbations in the encoded features and limiting robustness.

In contrast, pre-training leads to a markedly smoother and more structured representation space. Performances corresponding to different historical periods form coherent and compact clusters, indicating that stylistic characteristics are consistently captured at the encoder level. This separation is largely absent in the Supervised-only model, where Baroque, Classical, and Romantic samples are heavily entangled, reflecting weak style-specific structure and poor

*Table 2.* Robustness under transposition and tempo perturbation. For each test score, we generate five variants with random transposition (−6 to +5 semitones) and tempo scaling (0.7×–1.3×), and apply the same transformations to the ground-truth performances.

| Set | Vel JS | Vel Int | Dur JS | Dur Int | IOI JS | IOI Int | Ped JS | Ped Int | Overall JS | Overall Int |
|---|---|---|---|---|---|---|---|---|---|---|
| Original | 0.1805 | 0.8517 | **0.1879** | **0.8303** | 0.1740 | 0.8292 | **0.1111** | **0.8893** | 0.1634 | 0.8501 |
| Variant | **0.1455** | **0.8854** | 0.1952 | 0.8157 | **0.1569** | **0.8525** | 0.1181 | 0.8859 | **0.1539** | **0.8599** |

style adaptability.

More notably, the clusters in the pre-trained representation space are not arbitrarily separated but arranged in a smooth and ordered progression that mirrors the historical evolution of Western music, from Baroque through Classical and Romantic to Impressionist styles. This organization emerges without any explicit style supervision during training. The presence of such a structured continuum suggests that large-scale self-supervised pre-training enables the model to internalize fundamental musical regularities, such as harmonic organization and textural density, which vary smoothly across stylistic contexts and provide a stable representational foundation for performance rendering.

### 4.5. Robustness to Transposition and Tempo Perturbation

To examine whether the gains from pre-training come from memorizing specific pieces rather than learning a generalizable score-to-performance mapping, we conduct an additional robustness experiment on the test set. For each test score, we generate five perturbed variants by applying random transposition from −6 to +5 semitones and tempo scaling from 0.7× to 1.3×. The same transformations are applied to the corresponding ground-truth performances before evaluation, so that the generated and reference performances remain comparable.

As shown in Table 2, Pianist Transformer maintains comparable performance under these perturbations. The overall JS divergence slightly decreases from 0.1634 to 0.1539, and the overall intersection area increases from 0.8501 to 0.8599. These results suggest that the model does not simply memorize specific test pieces during pre-training, but learns a generalizable mapping from musical context to expressive performance.

### 4.6. Analysis of Scaling Effects and Architecture

In our final analysis, we conduct a preliminary exploration of scaling effects to understand the relationship between performance, scale, and our architectural choices. The results, presented in Figure 7, validate that our framework is scalable while also revealing potential bottlenecks that may inform future architectural design choices.

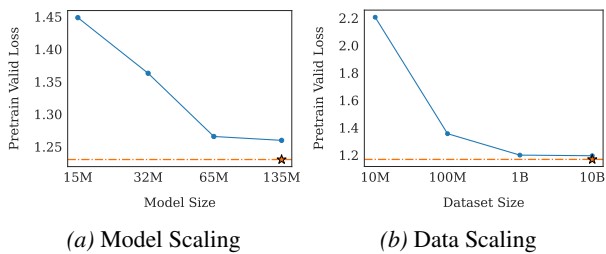

*(a)* Model Scaling  *(b)* Data Scaling

*Figure 7.* **A Preliminary Exploration of Scaling Effects.** Pretraining validation loss as a function of model size and data volume. (a) Increasing parameters in our asymmetric 10-2 architecture consistently improves performance, though saturation is observed at 135M. The star marks the lower loss achieved by 6-6 variant, suggesting the decoder as a bottleneck. (b) Increasing data yields substantial gains up to 1B tokens, after which performance plateaus, suggesting the 135M model's capacity becomes a limiting factor.

**Model Scaling and Decoder Bottleneck.** We first analyze the effect of model size. As shown in Figure 7a, increasing model parameters from 15M to 65M yields substantial performance gains. However, the curve flattens significantly from 65M to 135M, indicating performance saturation. We hypothesized that this bottleneck stems from our lightweight 2-layer decoder. To test this, we trained a symmetric 6-layer encoder, 6-layer decoder (6-6) variant with a more powerful decoder (marked by a star). It achieved a notably lower loss of 1.230 compared to our 135M model's 1.260, confirming that the shallow decoder is indeed the potential bottleneck for model capacity scaling.

**Data Scaling and Model Capacity Bottleneck.** Next, we examine the impact of data scale. Figure 7b shows a dramatic drop in loss when scaling data up to 1B tokens. However, the performance again saturates when increasing the data tenfold to 10B (1.199 vs. 1.195). To determine if this was also a decoder issue, we leveraged our more powerful 6-6 model. Even with this stronger architecture, the loss on 10B data only marginally decreased to 1.168. This suggests that for a massive 10B-token dataset, the overall model capacity (around 135M parameters) itself becomes the primary bottleneck, regardless of the encoder-decoder layer allocation.

**Architectural Trade-off.** Although the symmetric 6-6 architecture attains a lower pre-training loss, indicating that decoder depth limits capacity utilization during likelihood

*Table 3.* Quality and efficiency comparison between the symmetric 6-6 and asymmetric 10-2 architectures. Rendering quality is evaluated using overall JS divergence and intersection area. Efficiency metrics are reported relative to the 6-6 baseline.

| Model | JS Div. ↓ | Inter. ↑ | CPU Infer. | Train VRAM |
|---|---|---|---|---|
| 6-6 | 0.174 | 0.845 | 1.0× | 1.0× |
| 10-2 | 0.163 | 0.850 | 2.1× | 0.6× |

optimization, this advantage does not carry over to the final expressive rendering task. As shown by the objective evaluation on the test set, the asymmetric 10-2 model achieves comparable or slightly improved overall rendering quality, despite employing a substantially shallower decoder. This observation suggests that, under the current training regime and data scale, expressive performance rendering is primarily constrained by the quality of encoder representations rather than decoder depth.

In contrast, the efficiency benefits of the asymmetric design are pronounced. As summarized in Table 3, the 10-2 architecture delivers over a twofold improvement in CPU inference speed, reduces training-time VRAM consumption by approximately 40%, and increases training throughput by around 70% relative to the symmetric baseline. Taken together, these results indicate that the asymmetric 10-2 configuration offers a more favorable balance between rendering quality and computational efficiency, making it a practical choice for large-scale training and real-world deployment. A more detailed comparison across individual expressive dimensions and additional efficiency metrics is provided in Appendix F.

## 5. Conclusion

In this work, we introduced Pianist Transformer, demonstrating a shift in expressive piano performance rendering from a supervised paradigm to large-scale self-supervised learning. By learning from a 10B-token MIDI corpus under a unified representation, our approach overcomes the data limitations that have constrained prior methods in modeling the complex relationship between musical structure and expressive performance. Our experiments demonstrate that Pianist Transformer establishes a new state of the art across both objective metrics and subjective evaluations. In particular, subjective listening study shows that the model achieves human-level expressive quality under blind evaluation. Moreover, the model shows promising performance across diverse musical styles, suggesting the potential of large-scale self-supervised pre-training to improve stylistic robustness. Overall, our results indicate that scaling self-supervised learning provides a powerful foundation for music modeling, opening a new direction for future research in expressive piano performance rendering.

## 6. Limitations and Future Work

While Pianist Transformer demonstrates strong performance, it has several limitations that suggest future directions. First, although our robustness analysis suggests that the model does not simply memorize specific test pieces, some overlap between the in-the-wild pre-training corpus and the evaluation repertoire may still exist, which calls for future evaluation on newly composed or carefully deduplicated scores. Second, our efficient lightweight decoder remains a performance bottleneck for scaling, motivating research into more powerful yet efficient decoder architectures. Third, our focus on solo piano invites extending our self-supervised paradigm to multi-instrument and orchestral settings. Finally, the current system lacks explicit high-level control interfaces, such as global tempo control or natural-language conditioning, motivating future work on controllable rendering from more intuitive user inputs.

## Acknowledgments

This research was supported by the Jiangsu Science Foundation (BG2024036, BK20243012, BK20232003), Natural Science Foundation of China (62576162), the Fundamental Research Funds for the Central Universities (022114380023) and the "111 Center" (No. B26023).

The authors would like to thank Hao-Tian Chai and Huiyu Yi for their insightful and helpful discussions.

## Impact Statement

The pre-training and fine-tuning of our model were conducted only using publicly available datasets, as detailed in Appendix A. Our research did not involve the collection of new private data. For our subjective listening study, we recruited human participants. All participants were presented with an informed consent form prior to the study, which outlined the purpose of the research, the nature of the task, and how their data would be used. To protect participant privacy, all experimental responses were fully anonymized prior to analysis and were stored separately from any personal information required for compensation. We compensated each participant for their time and effort with a payment that exceeds the local minimum wage standard.

Pianist Transformer is intended as an assistive tool for composers, producers, and musicians to obtain expressive renderings of symbolic scores and to support music production workflows. Rather than replacing human performers, the system targets scenarios where editable MIDI renderings are useful for composition, previewing, arrangement, and creative exploration. We foresee no direct negative societal impacts resulting from this work, which is intended to advance research in computational music and creativity.

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

# A. Experimental Setup & Implementation Details

## A.1. Dataset Details

The pre-training corpus is constructed from the following sources. We applied specific preprocessing steps to ensure data quality and diversity.

- **Aria-MIDI** (Bradshaw & Colton, 2025): A large collection of over 1.1 million MIDI files transcribed from solo piano recordings. To ensure high fidelity, we only included segments with a transcription quality score above 0.95. Since the original transcriptions are relatively coarsely quantized, we applied random local augmentations to better simulate performance nuances and improve token coverage during pre-training. Specifically, we uniformly perturbed velocity values within a small local range and applied local random perturbations to duration and IOI values. These augmentations help expose the model to timing and velocity tokens that are otherwise underrepresented due to coarse quantization, while preserving the overall musical structure of the transcribed performances.

- **GiantMIDI-Piano** (Kong et al., 2022): A dataset of over 10,000 unique classical piano works transcribed from live human performances using a high-resolution system. These files retain fine-grained expressive details, including velocity, timing, and pedal events.

- **PDMX** (Long et al., 2025): A diverse dataset of over 250,000 musical scores, originally in MusicXML format. We used their MIDI conversions to provide our model with clean, score-based MIDI data. To filter out overly simplistic or empty files, we only included MIDI files larger than 7 KB.

- **POP909** (Wang et al., 2020): A dataset of 909 popular songs. We extracted the piano accompaniment tracks to include non-classical and accompaniment-style patterns.

- **Pianist8** (Chou et al., 2021): A collection of 411 pieces from 8 distinct artists, consisting of audio recordings paired with machine-transcribed MIDI files.

For SFT and evaluation, we use the **ASAP dataset** (Foscarin et al., 2020), a collection of aligned score-performance pairs of classical piano music.

Before training, we first normalize all MIDI files so they can be tokenized under a consistent format. For multi-track scores, we merge all tracks into a single time-ordered event stream and remove duplicate notes created during merging. We also convert every file to a fixed tempo of 120 BPM and rescale all onset times and durations. After this processing, all MIDI files share the same temporal scale and event structure, allowing reliable and uniform tokenization across the entire corpus.

To ensure precise note-level correspondence between scores and performances, we refined the provided alignments. We first employed an HMM-based note alignment tool (Nakamura et al., 2017) to establish a direct mapping for each note. For localized mismatches where a few notes could not be paired, we applied an interpolation algorithm to infer the correct alignment based on the surrounding context. Finally, segments with large, contiguous blocks of unaligned notes were filtered out and excluded from our training and evaluation sets to maintain high data quality. We create a strict piece-wise split by randomly holding out 10% of the pieces for our test set. The remaining 90% are used for fine-tuning.

## A.2. Model Architecture and Tokenizer

### A.2.1. MODEL ARCHITECTURE

Our Pianist Transformer employs an asymmetric encoder-decoder architecture based on the T5-Gemma framework (Zhang et al., 2025). The encoder is designed to be substantially deeper than the decoder, with 10 layers, to efficiently process long input sequences and build a rich contextual representation. The decoder, with only 2 layers, is lightweight to ensure fast and efficient autoregressive generation during inference. This design strikes a balance between expressive power and practical utility. Key hyperparameters for our 135M model are detailed in Table 4.

### A.2.2. UNIFIED MIDI REPRESENTATION

Central to our approach is a unified, event-based token representation that treats both score and performance MIDI identically. Each musical note is represented as a fixed-length sequence of eight tokens, capturing its core attributes and nuanced pedal

*Table 4.* Key hyperparameters for the Pianist Transformer.

| Parameter | Value |
| --- | --- |
| Model Architecture | T5-Gemma |
| Total Parameters | $\approx 135M$ |
| Hidden Size | 768 |
| Intermediate Size (FFN) | 3072 |
| Number of Encoder Layers | 10 |
| Number of Decoder Layers | 2 |
| Attention Head Dimension | 128 |
| Total Vocabulary Size | 5389 |

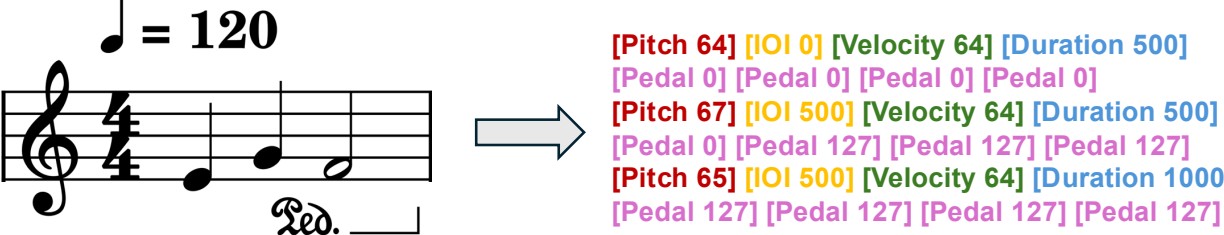

*Figure 8.* An Example of the Unified MIDI Representation

information. The sequence order is: `[Pitch, IOI, Velocity, Duration, Pedal1, Pedal2, Pedal3, Pedal4]`.

The vocabulary is structured as follows:

- **Pitch**: MIDI pitch values are mapped directly to 128 tokens (range 0 to 127).

- **Velocity**: MIDI velocity values are mapped to 128 tokens (range 0 to 127).

- **Timing (IOI & Duration)**: The Inter-Onset Interval (IOI) and Duration are quantized at a 1ms resolution and share a common vocabulary of 5000 tokens. The Duration can utilize the full range (0 to 4999), while the IOI is restricted to a slightly smaller range (0 to 4990) to avoid a known artifact in transcribed MIDI where durations frequently saturate at the maximum value.

- **Pedal**: Four pedal tokens represent the sustain pedal state sampled at four equidistant points within the interval leading to the next note. Each pedal value is mapped to one of 128 tokens (range 0 to 127). While this representation supports continuous (half-pedal) values, our pre-training data predominantly contained binary pedal events (0 or 127), effectively training the model to generate on/off pedal control.

Special tokens including `[PAD]`, `[MASK]`, `[BOS]`, `[EOS]`, and a special `[PLAY]` unused, resulting in a total vocabulary size of 5389. Figure 8 provides a concrete example of how a short musical phrase, under 120 BPM, is mapped into our 8-token event representation capturing pitch, timing, velocity, and pedal states.

### A.2.3. COMPARISON WITH PRIOR MIDI TOKENIZATION SCHEMES

Table 5 provides a comparison between our MIDI representation and several prior tokenization schemes. The design of our representation is guided by the specific requirements of the expressive performance rendering task, particularly the need to leverage large-scale MIDI data without structural features.

The core advantages of our representation are as follows:

- **Unlocks Self-Supervised Pre-training.** By using time shift for temporal representation, our method does not require structural features like bars or beats. This is a crucial advantage because it allows us to pre-train on massive datasets of performance MIDI, which lack this structure and thus cannot be used by other methods.

*Table 5.* Comparison of MIDI tokenization schemes. Our representation is tailored for scalable, self-supervised expressive performance rendering.

| Representation | Temporal Representation | Note-centric | Encodes Pedal |
|---|:---:|:---:|:---:|
| REMI (Huang & Yang, 2020) | Bar & Pos | No | No |
| MIDI-Like (Oore et al., 2020) | Time shift | No | No |
| CPWord (Hsiao et al., 2021) | Bar & Pos | No | No |
| Octuple (Zeng et al., 2021) | Bar & Pos | Yes | No |
| **Ours** | **Time shift** | **Yes** | **Yes** |

- **Designed for Performance Rendering.** Our note-centric approach treats each note and its properties as a single unit. This design is a natural fit for the rendering task, as it simplifies the process of matching an input score to an output performance and makes our model highly efficient.

- **Captures Essential Piano Acoustics.** Our representation explicitly encodes the sustain pedal, a crucial factor in producing the rich resonance characteristic of real piano performances. Incorporating pedal information enables the model to generate more expressive and realistic renderings.

This design keeps our representation simple and broadly compatible with different MIDI sources. While our setting focuses on symbolic music, related work in other long-horizon sequential domains has also explored using structured representations to improve generalization from raw experience (Shao et al., 2026).

### A.3. Training Procedure

#### A.3.1. SELF-SUPERVISED PRE-TRAINING

The pre-training phase is designed to build a foundational understanding of musical structure and expression from our large-scale, unlabeled MIDI corpus. We employ a masked denoising objective, similar to T5, where the model learns to reconstruct corrupted segments of the input token sequences. The choice of a pre-training objective is important because the utility of unlabeled data depends on the compatibility between the pretext task and the downstream target task (Jia et al., 2026a). In this stage, following recent mature practices in NLP for masked denoising objectives (Warner et al., 2025) we adopt a masking ratio of 0.3 with tokens randomly masked. The model was trained for 40,000 steps using the AdamW optimizer (Loshchilov & Hutter, 2019). Key hyperparameters for this stage are detailed in Table 6.

#### A.3.2. SUPERVISED FINE-TUNING

The fine-tuning process ran for 2 epochs. We adopted a slightly higher learning rate than in pre-training, which empirically led to faster convergence and a lower final loss. The learning rate followed a cosine decay schedule without a warmup phase. The global batch size was set to 32. All other settings remained consistent with the pre-training stage. A side-by-side comparison of pre-training and SFT hyperparameters is provided in Table 6.

*Table 6.* Comparison of hyperparameters for Pre-training and SFT stages.

| Parameter | Pre-training | SFT |
|---|:---:|:---:|
| Optimizer | AdamW | |
| Learning Rate Schedule | Cosine decay | |
| Peak Learning Rate | 3e-4 | 5e-4 |
| Warmup Steps | 2,500 | 0 |
| Training Duration | 40,000 steps | 2 epochs |
| Global Batch Size | 64 | 32 |
| Maximum Sequence Length | 4096 | |
| Precision | bfloat16 | |
| Hardware | 4x NVIDIA A800 GPUs | |

## A.4. Calculation of Objective Metrics

To measure how closely the generated performances resemble human playing, we compare the global token distributions of the model outputs with those of the human performances across the entire test set. For structured and underexplored modalities, carefully designed evaluation protocols are important for making model comparisons meaningful, as recent benchmark studies have also emphasized in other multimodal settings (Jia et al., 2026b). Accordingly, our objective metrics focus on distributional similarity across the key expressive dimensions of symbolic piano performance. For Velocity, Duration, and IOI, we aggregate the tokens of each type from all generated pieces and compute their distributions, which we then compare with the corresponding human distributions using JS Divergence and Intersection Area. For Pedal, where the corpus mainly contains binary values, we binarize both model outputs and human data and evaluate the distribution of the 16 possible joint configurations formed by the four pedal tokens of each note. For robustness, IOI and Duration statistics are computed within the ranges of 0-200 ms and 0-500 ms, which correspond to the high-frequency regions of the distributions and together cover over 90% of note events in the test set. This restriction mitigates evaluation bias introduced by heterogeneous time quantization across models, which can lead to systematic data absence in the long-tail regions and distort distributional comparisons.

To obtain a human baseline, for every piece, we treat one human performance as the candidate and use the remaining human performances as the reference set, applying the same distributional comparison. This provides a measure of the natural stylistic variation among human performers.

## A.5. Baseline Implementation

For VirtuosoNet-HAN and VirtuosoNet-ISGN, we used the official implementations and pre-trained weights, followed their recommended inference procedures, and selected the composer-style configurations that best matched the pieces in our test set.

ScorePerformer was evaluated under the same score-only setting. Since it is designed to operate with fine-grained style vectors derived from reference performances, which are not available in our setup, we adopted the unconditional generation mode recommended in the original paper, where style vectors are sampled from the prior distribution.

All generated MIDI files from all models were rendered to audio using the same high-quality piano soundfont to ensure a fair subjective listening study.

# B. Expressive Tempo Mapping Algorithm

To make our model's output compatible with standard music production software, we introduce the Expressive Tempo Mapping algorithm. This process converts the generated performance, which has timing in absolute milliseconds, into a standard MIDI file where expressive timing is encoded as a dynamic tempo map. This makes the performance fully editable within any DAW. The procedure is outlined in Algorithm 1.

The algorithm executes in three main stages:

1. **Tempo Estimation (Line 4):** First, we compare the timing of note onsets between the score MIDI ($M_{\text{score}}$) and the generated performance MIDI ($M_{\text{perf}}$). The differences in timing are used to calculate a local tempo (BPM) for each segment of the piece. This sequence of tempo changes forms a dynamic tempo curve, $T_{\text{changes}}$, which captures all the expressive timing (rubato) of the performance.

2. **Event Remapping (Lines 6-18):** Next, we create a new set of notes and pedal events. Each new note uses the pitch from the original score and the velocity from the generated performance. The crucial step is converting the onset time and duration of every note and pedal event from absolute milliseconds into musical ticks. This conversion is done using the tempo curve $T_{\text{changes}}$ estimated in the previous step. This aligns all events to a musical grid while preserving their expressive timing.

3. **Final Assembly (Line 19):** Finally, the newly created tempo curve ($T_{\text{changes}}$), the remapped notes ($N_{\text{aligned}}$), and the remapped pedal events ($CC_{\text{aligned}}$) are combined into a single, standard MIDI file ($M_{\text{DAW}}$). The resulting file sounds identical to the original performance but is now fully editable in a DAW, with all timing nuances represented in the tempo track.

---

**Algorithm 1** Expressive Tempo Mapping

---

1: **Input:** Score MIDI $M_{score}$, Performance MIDI $M_{perf}$
2: **Output:** DAW-friendly expressive MIDI $M_{DAW}$
3: Extract notes $N_{score}, N_{perf}$ and pedal events $CC_{perf}$ from input files.
4: Estimate a dynamic tempo curve $T_{changes}$ based on timing deviations.
5: Initialize empty lists for aligned events: $N_{aligned}, CC_{aligned}$.
6: **for** each corresponding note pair $(n_{score}, n_{perf})$ **do**
7:     Create a new note $n_{new}$ where:
8:         - pitch is from pitch of $n_{score}$
9:         - velocity is from velocity of $n_{perf}$
10:         - onset in ticks is converted from $n_{perf}$'s onset in milliseconds using $T_{changes}$.
11:         - duration in ticks is converted from $n_{perf}$'s duration in milliseconds using $T_{changes}$.
12:     Append $n_{new}$ to $N_{aligned}$.
13: **end for**
14: **for** each control event $cc$ in $CC_{perf}$ **do**
15:     Convert $cc$'s timestamp from milliseconds to ticks using $T_{changes}$ to get $t_{new}$.
16:     Create a new control event $cc_{new}$ with value from $cc$ and time from $t_{new}$.
17:     Append $cc_{new}$ to $CC_{aligned}$.
18: **end for**
19: Assemble $M_{DAW}$ by combining $T_{changes}$, $N_{aligned}$, and $CC_{aligned}$.
20: **return** $M_{DAW}$

---

## C. Subjective Listening Study Details

To conduct a definitive, human-centric evaluation of our model's performance, we designed and carried out a comprehensive subjective listening study. This appendix provides a detailed account of the study's design, participants, materials, and procedures.

### C.1. Participant Demographics

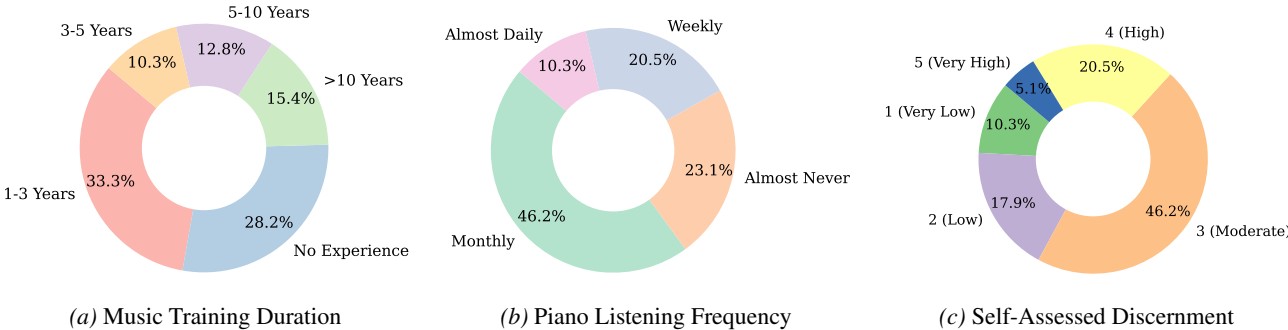

*(a)* Music Training Duration          *(b)* Piano Listening Frequency          *(c)* Self-Assessed Discernment

*Figure 9.* **Demographic distribution of the 39 participants in the listening study.** The plots show (a) the duration of formal music training, (b) the frequency of listening to classical piano music, and (c) self-assessed ability to discern piano music quality on a 1-5 scale. This diverse composition validates the generalizability of our study's findings.

Our subjective listening study's validity rests on the quality and diversity of its participant pool. We initially recruited 57 individuals; after a rigorous screening for attentiveness and completion quality, 39 responses were retained for the final analysis. To mitigate fatigue effects and maintain reliable ratings, we kept the listening task within a reasonable duration rather than further expanding the evaluation scale. This section details the demographic composition of this group, providing evidence for its suitability for the nuanced task of evaluating musical expression.

The detailed distributions of participants' musical experience and listening habits are visualized in Figure 9. Several key characteristics of the group bolster the credibility of our findings:

**Balanced Expertise Spectrum (Figure 9a).** The participants' formal music training is not skewed towards one extreme. The pool includes a substantial proportion of listeners with no formal training (28.2%), ensuring that our model's appeal is not limited to musically educated ears. Concurrently, the presence of highly experienced individuals (15.4% with > 10 years of training) guarantees that subtle expressive details are also being critically evaluated. This heterogeneity mitigates potential bias and strengthens the generalizability of our preference results.

**Representative Listening Habits (Figure 9b).** The distribution of classical piano listening frequency reflects a general audience rather than a niche group of connoisseurs. The largest segment listens "Monthly" (46.2%), suggesting that the superior performance of Pianist Transformer is perceptible and appreciated even by those who are not deeply immersed in the genre daily.

**Competent and Calibrated Self-Assessment (Figure 9c).** The self-assessed ability to discern music quality is centered around "Moderate" (46.2%), with a healthy portion rating themselves as "High" (20.5%). This distribution suggests a group that is confident in their judgments without being overconfident, indicating that the participants were well-suited for the evaluation task.

In summary, the participant pool is intentionally diverse, comprising a mix of novices, enthusiasts, and experts. This composition ensures that our findings are robust, reliable, and reflective of a broad range of listener perceptions.

## C.2. Music Clips for Evaluation

The listening study was based on six music clips, each approximately 15 seconds long. To ensure an unbiased comparison, all clips were systematically taken from the beginning of each piece. The selection was also deliberately curated for stylistic breadth, featuring works from the Baroque, Classical, and Romantic periods, as well as modern pop style. This diversity provides a rigorous testbed for evaluating the models' generalization abilities across varied musical contexts. The specific pieces are detailed in Table 7.

*Table 7.* Musical excerpts selected for the subjective listening study, highlighting their stylistic diversity.

| Composer | Work | Period / Style |
| --- | --- | --- |
| J. S. Bach | Prelude and Fugue in G minor, BWV 885, Prelude | Baroque |
| J. Haydn | Keyboard Sonata No. 58 in C major, Hob.XVI:48:II | Classical |
| L. v. Beethoven | Piano Sonata No. 4 in E-flat major, Op. 7:I | Classical |
| F. Chopin | Étude in D-flat major, Op. 25, No. 8 | Romantic |
| F. Liszt | Étude d'exécution transcendante No. 1, Preludio, S. 139 | Romantic |
| Joe Hisaishi | Merry-Go-Round of Life | Modern Pop |

## C.3. Case Study: Generalization to Out-of-Domain Popular Music

To further probe the generalization capabilities of our model, we included a musical excerpt from a modern popular song. Robustness under distributional shifts has been widely discussed in recent foundation-model systems, where models trained in relatively static settings may degrade when deployed in changed environments (Wu et al., 2026). In our setting, this piece is stylistically distinct from the primarily classical ASAP dataset used for fine-tuning, thereby serving as a challenging out-of-domain test. The goal was to assess whether the musical understanding gained during pre-training would translate effectively to genres beyond the immediate scope of the fine-tuning data.

The results of this case study, summarized in Figure 10, reveal a nuanced dynamic. We observe that VirtuosoNet-ISGN delivers a highly competitive performance on this slow, lyrical piece. Its multi-dimensional ratings (Figure 10a) and average rank (Figure 10b) are nearly on par with our Pianist Transformer, suggesting that the expressive patterns it learned are well-suited for this particular style of song-like playing.

However, a crucial distinction emerges from the first-place vote rate (Figure 10c). Despite the close average scores, listeners chose Pianist Transformer as the single best performance by a dominant margin. This finding highlights a key advantage of our approach. While other systems may produce competent or even good performances on stylistically favorable pieces, Pianist Transformer is significantly more likely to generate a truly exceptional rendering that listeners perceive as the

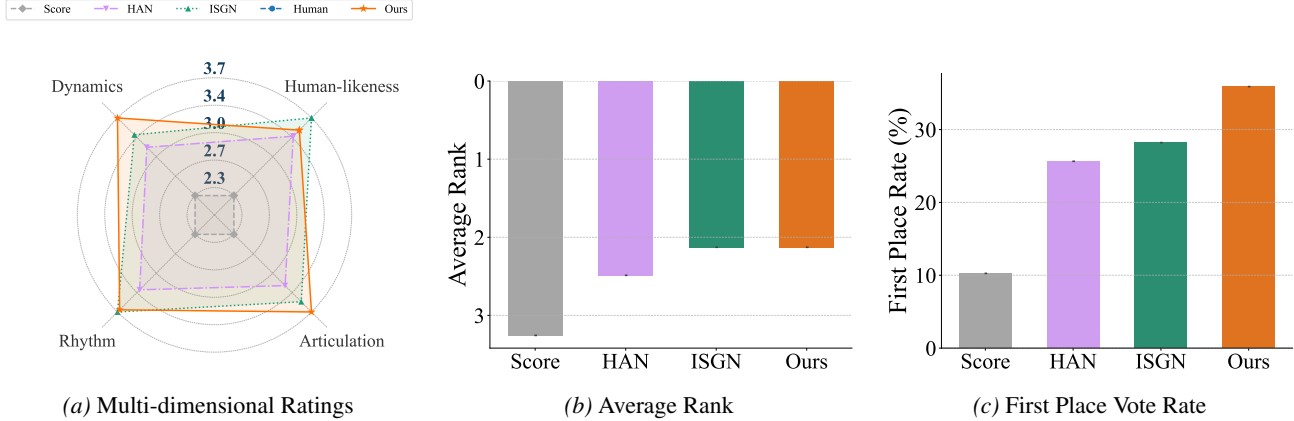

*(a)* Multi-dimensional Ratings      *(b)* Average Rank      *(c)* First Place Vote Rate

*Figure 10.* **Case Study on an Out-of-Domain Popular Music Excerpt.** Subjective evaluation results for a slow, lyrical pop piece. (a, b) VirtuosoNet-ISGN performs competitively in average ratings and rankings. (c) However, our **Pianist Transformer** secures a dominant share of first-place votes, indicating superior overall appeal and quality.

definitive best. We attribute this advantage to the fine-grained expressive cues learned during large-scale pre-training, which may help the model produce more natural and preferred musical interpretations beyond the fine-tuning domain. Exploring test-time adaptation for personalized or style-specific rendering is an interesting direction for future work (Zhou et al., 2026).

This case study provides preliminary evidence that Pianist Transformer can generalize to out-of-domain popular music, but the current evaluation remains centered on classical solo piano. Since the supervised fine-tuning data mainly consists of classical score-performance pairs, the model may inherit style biases in timing, phrasing, articulation, and tempo fluctuation. This may be particularly important for genres with more flexible or genre-specific rhythmic conventions, such as jazz and modern popular music. Therefore, more systematic evaluation on pop, jazz, and other modern styles is needed to better characterize the model's cross-genre generalization ability.

### C.4. Reliability and Consistency Analysis

To ensure the robustness and impartiality of our subjective evaluation, we embedded an internal consistency check within the study. For one musical excerpt (Liszt's work), two identical audio clips from the same model's performance were presented to each participant as if they were distinct versions. The analysis of ratings for these duplicates, shown in Table 8, provides crucial insights into the study's validity.

First, the analysis confirms the experiment's impartiality. The Mean Error (Bias) between the ratings for the duplicate clips is negligible, and a paired t-test showed these differences to be statistically insignificant (all $p > 0.6$). This result demonstrates that our experimental design successfully mitigated systematic biases, such as those arising from presentation order or listener fatigue. Furthermore, the Pearson correlation ($r$) between the paired ratings is positive but modest. This is an expected outcome, reflecting the inherent variability and noise in the subjective human perception of music. The presence of this natural perceptual uncertainty makes our main findings, the clear and statistically significant preference for Pianist Transformer, even more compelling. It indicates that the perceived quality difference between our model and its counterparts was strong and consistent enough to overcome this noise, thereby solidifying the significance and reliability of our conclusions.

*Table 8.* Intra-rater reliability analysis on duplicate audio stimuli. We report the Mean Error (Bias) and Pearson Correlation ($r$) between ratings for two identical audio clips presented to the same user. The low, statistically non-significant bias confirms the experiment's impartiality.

| Metric | Dynamics | Rhythm | Articulation | Human-likeness |
|---|---|---|---|---|
| Mean Error (Bias) | 0.026 | -0.103 | 0.000 | 0.051 |
| Pearson Corr. ($r$) | 0.155 | 0.377 | 0.351 | 0.133 |

## D. Ablation Study of the Masking Ratio in Pre-training

To analyze the effect of the masking ratio on downstream performance, we conducted an ablation study by pre-training models with three different ratios: 15%, 30% and 45% and then fine-tuning them on our rendering task. The results are presented in Table 9.

The results indicate that our main setting of 30% outperforms the 15% ratio, while the 45% ratio yields slightly better overall performance. This suggests that the optimal masking strategy for symbolic music may differ from common practices in NLP, an interesting direction for future work.

However, all three pre-trained models significantly outperform the supervised-only baselines, showing that the gains from self-supervised pre-training do not depend on any specific masking setting.

*Table 9.* Ablation study on the pre-training masking ratio. While the 45% ratio achieves the best overall scores, all pre-trained variants significantly outperform supervised-only baselines.

| Mask Ratio | Velocity | | Duration | | IOI | | Pedal | | Overall (Avg.) | |
|---|---|---|---|---|---|---|---|---|---|---|
| | JS Div (↓) | Inter. (↑) | JS Div (↓) | Inter. (↑) | JS Div (↓) | Inter. (↑) | JS Div (↓) | Inter. (↑) | JS Div (↓) | Inter. (↑) |
| 0.15 | 0.2127 | 0.8087 | 0.1801 | 0.8359 | 0.1882 | 0.8145 | 0.1364 | 0.8590 | 0.1794 | 0.8295 |
| 0.30 | 0.1805 | 0.8517 | 0.1879 | 0.8303 | **0.1740** | **0.8292** | **0.1111** | **0.8893** | 0.1634 | 0.8501 |
| 0.45 | **0.1393** | **0.8941** | **0.1774** | **0.8414** | 0.1816 | 0.8211 | 0.1135 | 0.8826 | **0.1530** | **0.8598** |

## E. Efficiency Analysis of Sequence Compression and Asymmetric Architecture

To investigate how our efficiency-related components influence the overall training efficiency, we conduct a detailed analysis of note-level sequence compression and the asymmetric architecture. Each component individually reduces computational cost, but their combination leads to a substantially larger improvement than using either one alone.

*Table 10.* Synergistic Efficiency Analysis of Sequence Compression and the Asymmetric Architecture. We report relative metrics where the baseline (6-6, Uncompressed) is set to 1.00x. Lower is better for VRAM, higher is better for Speed.

| Representation | Metric | Architecture | |
|---|---|---|---|
| | | 6-6 | 10-2 |
| Uncompressed | Training VRAM | 1.00x | 0.89x |
| | Training Speed | 1.00x | 1.07x |
| Compressed | Training VRAM | 0.63x | **0.38x** |
| | Training Speed | 1.81x | **3.13x** |

As shown in table 10, compression accelerates training by 1.81× on the 6–6 architecture, and the 10–2 architecture improves speed by 1.07× without compression. However, when both components are applied together, the overall training speed reaches 3.13×, substantially exceeding the product of their individual gains. A similar synergistic effect appears in VRAM reduction, where compression and architectural asymmetry jointly amplify memory savings.

These findings confirm that the proposed efficiency components are not merely additive but interact in a way that significantly amplifies their benefits—effectively achieving a "$1 + 1 > 2$" efficiency outcome.

## F. Detailed Discussion about Architectural Trade-off.

We chose an asymmetric 10-2 architecture to balance performance and efficiency. To validate this choice, we compare it against a symmetric 6-6 baseline with a similar parameter count.

Table 11 shows the final rendering performance of both models after fine-tuning. While the symmetric 6-6 model achieves a slightly lower loss during pre-training, this does not translate to superior performance on the downstream rendering task. Our 10-2 model performs comparably to the 6-6 variant.

While the final rendering quality is highly comparable between the two architectures, the choice is justified by the significant

*Table 11.* Objective evaluation results on the ASAP test set, comparing the asymmetric 10-2 architecture with a symmetric 6-6 baseline. Despite the 6-6 model having a deeper decoder, our 10-2 model achieves comparable or even slightly better performance on the final rendering task.

| Model | Velocity | | Duration | | IOI | | Pedal | | Overall (Avg.) | |
|---|---|---|---|---|---|---|---|---|---|---|
| | JS Div (↓) | Inter. (↑) | JS Div (↓) | Inter. (↑) | JS Div (↓) | Inter. (↑) | JS Div (↓) | Inter. (↑) | JS Div (↓) | Inter. (↑) |
| 6-6 | **0.1797** | 0.8411 | 0.2070 | **0.8427** | 0.1951 | 0.8116 | 0.1158 | 0.8837 | 0.1744 | 0.8448 |
| 10-2 | 0.1805 | **0.8517** | **0.1879** | 0.8303 | **0.1740** | **0.8292** | **0.1111** | **0.8893** | **0.1634** | **0.8501** |

gains in computational efficiency, detailed in Table 12. Our 10-2 model is over 2x faster in CPU inference and substantially more resource-efficient during training, requiring 40% less VRAM and achieving a 60% faster throughput.

*Table 12.* Efficiency comparison between the asymmetric (10-2) and symmetric (6-6) architectures. The 6-6 model is set as the baseline for relative speed and memory usage. Our design significantly accelerates both training and inference while reducing memory footprint.

| Metric | 10-2 | 6-6 | Advantage |
|---|---|---|---|
| CPU Inference Speed | 2.1x | 1.0x | 110% faster |
| Training VRAM | 0.6x | 1.0x | 40% less memory |
| Training Speed | 1.7x | 1.0x | 70% faster |

In conclusion, our asymmetric 10-2 architecture provides a superior trade-off, delivering state-of-the-art rendering quality while being significantly more efficient for both training and deployment. This makes it a more practical solution.

## G. Inference Strategy

During inference, our goal is to generate expressive performances that remain strictly matched to the input score. To preserve the note-level correspondence, we apply a hard pitch constraint: the model is free to sample expressive attributes but whenever a Pitch token is expected, we directly set it to the pitch from the input score instead of sampling. This enforces a one-to-one mapping between score notes and generated performance notes.

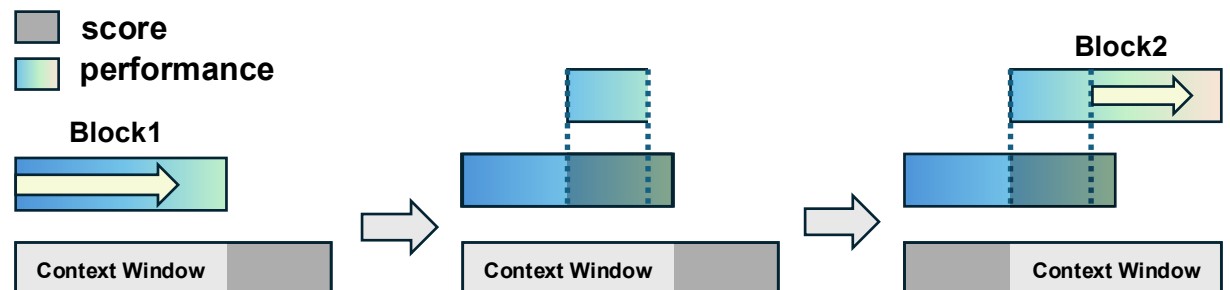

*Figure 11.* **Overlapped Block-wise Generation Strategy.** This figure illustrates how we generate long sequences using overlapping blocks. Block 1 is produced first. For the next block, we shift the window forward and reuse the stable overlapping region from the previous block as the decoder's context. Block 2 then continues generation from this context.

For pieces longer than 4096 tokens, we use an overlapped block-wise generation strategy illustrated in Figure 11. We first generate a 4096-token block. For the next block, we shift the window forward by 2048 tokens so that the new encoder input overlaps with the second half of the previous block. As decoder context, we reuse this overlapping region but drop a few unstable tokens at the tail before using it as the prompt. The model then continues generation from this context, and we append only the newly produced part. This procedure repeats until the piece is complete.

