# OpenReview forum: "Pianist Transformer: Towards Expressive Piano Performance Rendering via Scalable Self-Supervised Pre-Training"
_ICML.cc/2026/Conference — ICML 2026 regular_

### Official Review · Reviewer_b9Tf · 2026-03-01

**Soundness:** 3
**Presentation:** 4
**Significance:** 3
**Originality:** 3
**Overall Recommendation:** 5
**Confidence:** 4

**Summary:**

This paper deals with expressive piano music performance rendering. The authors devised a strategy that utilizes self-supervised learning to make use of a large corpora of unlabeled scores and MIDI performances, followed by a supervised fine tuning stage, to train their newly proposed model named Pianist Transformer. The proposed approach was thoroughly evaluated through both objective and subjective tests, achieving better performance compared to previous baselines, and reaching levels of human-like expressive performance (almost) comparable to those of real human performances. Additional ablation studies and visualisations further support the soundness of the presented approach.

**Compliance With Llm Reviewing Policy:**

Affirmed.

**Final Justification:**

My main concerns from the rebuttal were tied to the subjective evaluation, which used only a few different music clips, and the related stylistic robustness report, which was heavily influenced by this fact.

The authors have clarified their listening test design in detail, which had to take listener fatigue into account due to the total number of music samples (each of the 6 clips had 5 versions, resulting in 30 clips in total). Furthermore, the authors acknowledged the limitation in the stylistic robustness report and adjusted the claims accordingly.

As such, I am raising the final recommendation from "weak accept" to "accept".

**Key Questions For Authors:**

1. Given the small number of samples used for the listening study, would it be possible to expand the number of samples tested, especially to support the claims about Stylistic Robustness?

2. In Appendix A.1, you mentioned that Aria-MIDI dataset was slightly augmented in velocity, duration, and IOI. Given that this dataset represents the largest portion of your overall data, would it be possible to include specifics of these augmentations for better understanding and reproducibility?

**Limitations:**

Yes

**Strengths And Weaknesses:**

**Strengths:**
1) Writing feels smooth and polished.
2) Evaluation is quite rigorous with ablation studies and multiple visualisations supporting the feasibility of the proposed method. The results confirm the feasibility of the proposed approach as it outperforms existing baselines.
3) This work is the first to deploy self-supervised learning for expressive music performance rendering.

**Weaknesses:**
1) Subjective evaluation is built around rating 6 samples, which is quite a low number. Especially, when it comes to assessing the results by musical styles, each style is only represented by 1 or 2 samples, which is highly insufficient to arrive at generalised conclusions.

**Detailed review:**
1) Slight inconsistencies in typography – the bold key phrases starting paragraphs end with a dot for the most of the paper, but not at the end of Introduction where 2 out of 3 of these bold subtitles end with a colon. Please, unify the notation through the paper.

---

> ### Author Rebuttal · Authors · 2026-03-26
>
> We thank the reviewer for the positive feedback on the writing quality, evaluation, and ablation studies.  Below we respond to the reviewer’s suggestions and concerns:
> 1. **Limited number of subjective evaluation samples.** We agree that the scale of the listening study is relatively limited. The listening study includes 6 clips (each around 15 seconds), with 5 versions per clip (except for the out-of-domain pop example, which does not include a human performance). Attention-check items are also included. As a result, each listener evaluates around 30 clips. To avoid listener fatigue, we did not further expand the study. The study adopts randomized presentation and attention-check filtering, and we further provide a detailed reliability analysis in Appendix C.4. In addition, the subjective findings are consistent with the objective metrics computed on the full test set, which provide supporting evidence for our conclusions. We also agree that recruiting more participants and distributing the samples across groups to increase the scale of the subjective evaluation would help yield more reliable conclusions. However, given current resource constraints, we are unable to further expand the study at this stage and consider this an important direction for future exploration.
> 2. **Aria-MIDI augmentation details.** We agree that providing more details about Aria-MIDI augmentation would improve clarity and reproducibility. Our augmentation mainly applies uniformly random perturbations to coarse-grained velocity and timing within local ranges to ensure that tokens that are rare due to coarse quantization are sufficiently observed during pre-training. We will include these details in the camera-ready version.
> 3. **Formatting consistency.** This suggestion is entirely valid. We will unify the punctuation style after the three bold subtitles at the end of the Introduction and end them all with periods.
>
> We again thank the reviewer for the supportive comments and careful review.

---

> > ### Author Rebuttal · Reviewer_b9Tf · 2026-04-04
> >
> > Dear Authors,
> >
> > Thank you for addressing the points raised. Points 2 and 3 are fine with me now, but I have a follow-up questions/requests regarding point 1.
> >
> > The explanation provided for the number of samples in the subjective evaluation is valid, as I agree listener fatigue can be a complicating factor and should be avoided.
> >
> > Nevertheless, 5 versions per clip still represent the same musical piece, which impairs the results and conclusions of the Stylistic Robustness experiment. The reliability analysis in Appendix C.4, from my understanding, deals only with consistency of ratings, as it compares two identical versions of a single clip presented to the listeners as two different versions. However, this still does not cover for the low number of musical pieces (clips) present, especially in the Stylistic Robustness setting, where each “style” group contains only 1 or 2 clips. In a scenario where each stylistic group contained just 1 piece, could you still correctly refer to it as a stylistic group? (you are not far from this)
> >
> > It is not clear to me which objective evaluation results support the claims about Stylistic Robustness – I do not think any of them do. (The visualization in Figure 6 is nice and shows a continuum in the latent space, however, it is unrelated to the the subjective evaluation)
> >
> > Thus, I would suggest the authors to clearly state this low number of clips per style in their Stylistic Robustness paragraph (Section 4.3) and soften the language about the achieved results in it, for example, by editing the following part:
> >
> > >“In contrast, Pianist Transformer maintains consistently high human-likeness ratings across all styles. This stylistic robustness likely benefits from large-scale pre-training on diverse musical data, enabling the model to generalize beyond the stylistic biases of the fine-tuning dataset.“
> >
> > In the Conclusion Section, this should be altered too:
> > >“Moreover, the model exhibits strong robustness across diverse musical styles, highlighting the generalization benefits enabled by large-scale self-supervised pre-training.“
> >
> > The achieved results, given the low number of clips per style, do not fully demonstrate this robust behaviour as described, although, they could still suggest it. Therefore, I kindly ask the authors to propose relevant edits to these parts as well as acknowledging this limitation on number of music pieces in your limitations section mainly regarding the stylistic robustness analysis.
> >
> > Looking forward to your response,
> >
> > Reviewer b9Tf

---

> > > ### Author Response · Authors · 2026-04-04
> > >
> > > Thank you for further pointing this out. We agree with your main concern: the current evidence is not sufficient to support a strong claim that the model exhibits stylistic robustness. Instead, these results should be interpreted as preliminary positive evidence from a limited set of samples.
> > >
> > > We also agree that the reliability analysis in Appendix C.4 and the objective metrics in the paper mainly support the overall effectiveness of our method in performance rendering quality, and do not directly support a stronger conclusion about stylistic robustness. Thank you for making this distinction clear.
> > >
> > > Following your suggestion, we will revise the relevant discussion in Section 4.3, the Conclusion, and the limitations section to explicitly acknowledge the limited number of clips in each style group and to moderate the strength of our claims accordingly.
> > >
> > > Specifically, we plan to revise the stronger statement in Section 4.3 to:
> > > > In contrast, Pianist Transformer maintains high human-likeness ratings across the evaluated styles. While the number of clips in each style group is limited, these results suggest a potential benefit of large-scale pre-training on diverse musical data in improving the model's adaptability to different musical styles.”
> > >
> > > We will also revise the corresponding sentence in the Conclusion to:
> > > > Moreover, the model shows promising performance across diverse musical styles, suggesting the potential of large-scale self-supervised pre-training to improve stylistic robustness.”
> > >
> > > In addition, we will add to the limitations discussion that the current analysis of stylistic robustness is constrained by the small number of musical excerpts in each style group, and that a larger-scale study covering more pieces would be needed to draw firmer conclusions.
> > >
> > > Thank you again for this important suggestion. We will revise these claims accordingly and make the limitation explicit in the final version.

---

### Official Review · Reviewer_ab4i · 2026-03-08

**Soundness:** 1
**Presentation:** 2
**Significance:** 2
**Originality:** 2
**Overall Recommendation:** 2
**Confidence:** 5

**Summary:**

This paper applies large-scale self-supervised pre-training to expressive piano performance rendering (mapping a symbolic score to a human-like performance). The authors propose a unified MIDI tokenization for scores and performances, pre-train on 10B tokens of unlabeled MIDI, then fine-tune on ~100 hours of aligned score-performance pairs. They introduce an asymmetric encoder-decoder Transformer with note-level compression (8 tokens per note collapsed into one vector for the encoder) and a post-processing algorithm for DAW-compatible output. Results on the ASAP dataset show improvements over baselines on objective metrics and a subjective listening study.

**Compliance With Llm Reviewing Policy:**

Affirmed.

**Final Justification:**

My final recommendation for this submission leans negative (2 Reject).

My primary concern is that there is a fundamental flaw in the evaluation: the model has, with near certainty, seen _all_ of the pieces in the evaluation set during pre-training. The additional experiments on scores w/ shifted transposition or tempo do not eliminate this concern. E.g., the model could just be learning to apply analogous transformations to the performances of that same material seen at pre-training time. An evaluation on novel scores, or a deeper analysis of the similarity between generated performances and those from the pre-training data, would be needed to address this concern. The authors cite similar norms from other published papers (MASS/BART) - I would argue that two wrongs don't make a right.

A secondary concern is the limited relevance of this work to the broader ICML community. Ultimately, the core method of pre-training on domains A/B and then fine tuning on a small amount of paired (A->B) data is well known. As the authors point out in their rebuttal, there _is_ a modest degree of methodological novelty here, but most of it is specific to the music domain (unified score/perf representation, note-level compression, expressive tempo mapping). The parts that could be broader (asymmetric encoder-decoder) are neither the focus of the work nor evaluated in any other domain.

**Key Questions For Authors:**

1. Can you confirm that the ASAP test pieces are excluded from the pre-training corpus? If not, how do you distinguish genuine expressiveness from memorization?
1. How exactly are score-level attributes (metrical timing, categorical dynamics) mapped into the unified millisecond/velocity representation, and what information is lost?
1. Why was a denoising pre-training objective chosen over continuation, which might seem more natural for autoregressive performance generation?
1. Does the decoder generate all 8 tokens per note autoregressively? If so, the sequence compression only applies to the encoder side, and the efficiency claims should be qualified accordingly.
1. Why do the Human performances in Table 1 not achieve perfect scores, given that they presumably define the reference distribution?
1. Are there qualitative or quantitative results on music outside the classical domain (e.g., pop, jazz)? This would seem to be a natural advantage of large-scale pre-training and would help address concerns about memorization of ASAP pieces.

**Limitations:**

- Evaluation is likely contaminated by pre-training data overlap with ASAP test pieces, making all reported improvements unreliable.
- Core architectural and tokenization details are insufficiently specified in the main text.
- Applicability beyond classical music (the domain of ASAP) is not evaluated, despite being a natural advantage of large-scale pre-training.
- The impact statement claims "no direct negative societal impacts," which is an insufficient treatment of the potential effects of automating expressive musical performance — including displacement of working pianists, deskilling of performance practice, and training on performers'/composers' data at scale and without explicit consent.

**Strengths And Weaknesses:**

### Strengths

**Simple, well-motivated approach.** The idea of applying self-supervised pre-training to score-to-performance rendering is straightforward and addresses a real bottleneck (scarcity of aligned data) in a mainstay task in music AI.

**Reasonably convincing results.** Both quantitative metrics (Table 1) and the subjective listening study show clear improvements over baselines, with listeners unable to reliably distinguish the model's output from human performances.

**Interesting architectural finding.** The deep encoder / shallow decoder configuration achieving strong rendering quality with substantially improved efficiency (2.1× inference speedup, 0.6× VRAM) is a potentially broader finding worth noting, and the comparison with the symmetric 6-6 variant in Section 4.5 is appreciated.

### Weaknesses

**Likely data contamination in evaluation.** Despite a "strict piece-wise split" on ASAP, the test set pieces are almost certainly present in the 10B-token pre-training corpus of in-the-wild MIDI. This fundamentally undermines the evaluation — it is unclear whether the model's improved performances reflect genuine expressive understanding or memorized renditions of pieces seen during pre-training. The human evaluation should center on novel scores not present in the pre-training data.

**Underspecified tokenization and architecture.** Several core details are missing or deferred entirely to the appendix. How are scores (which use metrical timing and categorical dynamics) converted into the unified millisecond/velocity representation? How exactly are 8 token embeddings aggregated into a "consolidated vector"? And critically, how are those consolidated vectors modeled in a standard autoregressive setup? Does the decoder still generate all 8 tokens autoregressively while the encoder sees compressed single vectors? If so, the claimed sequence length reduction only applies to the encoder, which should be stated explicitly.

**Inadequate related work.** The paper omits substantial bodies of related work. The framing of "self-supervised learning in music" ignores the extensive lineage of autoregressive MIDI generation (PerformanceRNN, Music Transformer, MuseNet, Anticipatory Music Transformer), which is fundamentally self-supervised. Related work on compact note-level modeling (Compound Word Transformer, U-MusT) and expressive tempo mapping (e.g., REMI) is also missing.

### Minor concerns

**Abstract clarity.** The abstract is not written for a general ICML audience — "expressive music performance rendering" is ambiguous without clarification that this is conditional generation (score → performance), not unconditional generation.

**Unexplained human baseline scores.** The Human row in Table 1 shows imperfect metrics (JSD > 0, Intersection < 1). If human performances define the reference distribution, why don't they achieve perfect scores? This needs explanation.

**Mischaracterized efficiency claims.** The claim of reducing complexity from O(N²) to O((N/8)²) misuses asymptotic notation — this is still O(N²). The paper should instead argue the practical constant-factor improvement (fitting more context into fixed memory), ideally with concrete numbers.

**Pre-training data choices.** Why not also pre-train on the large quantity of available digital scores (e.g., PDMX)? Given the unified representation, this seems like a natural extension.

**Limited audience interest.** Score-to-performance rendering is important within music AI but may be of limited broader interest to the ICML community.

---

> ### Author Rebuttal · Authors · 2026-03-26
>
> We thank the reviewer for the detailed evaluation. We believe several concerns stem from misunderstandings of points already described in the paper, and we clarify them below.
> 1. **Pre-training data contamination.** Our pre-training objective is denoising reconstruction, which learns general musical representations (Figure 6 in paper) rather than score-to-performance mappings. There is no aligned supervision signal at this stage, and the input–output format differs fundamentally from the downstream task, so this does not constitute direct contamination. To address the concern that the model may rely on memorization, we conduct an additional experiment: for each test score, we generate 5 variants with random transposition (−6 to +5 semitones) and tempo scaling (0.7×–1.3×), and apply the same transformations to the ground truth. If the model relied on memorization, performance should degrade significantly under these perturbations. However, results show comparable (and sometimes slightly improved) quality, suggesting the model learns a general mapping rather than memorizing specific pieces, and that the evaluation remains valid.
>
> | Set   | Vel JS | Vel Int | Dur JS | Dur Int | IOI JS | IOI Int | Ped JS | Ped Int | Overall JS | Overall Int |
> |---------|--------|-----------|--------|-----------|--------|-----------|-----------|--------------|------------|----------------|
> | Origin  | 0.1805 | 0.8517    | **0.1879** | **0.8303**    | 0.1740 | 0.8292    | **0.1111**    | **0.8893**       | 0.1634     | 0.8501         |
> | Variant | **0.1455** | **0.8854**    | 0.1952 | 0.8157    | **0.1569** | **0.8525**    | 0.1181    | 0.8859       | **0.1539**     | **0.8599**         |
>
> 2. **Tokenization and representation details.** Our data format is MIDI, from which onset, duration, and velocity are standardly extracted. These details are already described in Appendix A.1 (normalization and timing recalibration) and Appendix A.2.2 (tokenization specifications with concrete examples).
> 3. **Architecture description and compression.** These design choices are already described in Sections 3.2–3.3 and further analyzed in Appendices E–F. Sequence compression is applied only on the encoder side, while the decoder remains autoregressive and intentionally shallow for efficiency.
> 4. **Pre-training objective.** Given our encoder–decoder architecture, masked denoising is a natural choice [1]. In contrast, continuation is generally more suitable for decoder-only models and may under-train the encoder in our setting.
> 5. **Related work.** While these works are relevant, they address different settings from ours. Prior autoregressive models (e.g., PerformanceRNN, Music Transformer) mainly focus on unconditional generation or continuation with single-stage training, whereas our work studies a pre-training + fine-tuning paradigm for structured conditional generation. We already include comparisons with representations such as REMI and CPWord in Table 4 (Appendix A.2.3), and will make this distinction clearer in the final version.
> 6. **Human baseline, PDMX, and out-of-domain results.** These points are already addressed in the paper. Appendix A.4 explains that the Human baseline reflects variation across different human performances, rather than an oracle score; Appendix A.1 includes PDMX as pre-training data; and Appendix C.3 provides an out-of-domain pop case study.
> 7. **Efficiency wording.** We agree that presenting O((N/8)^2) as a new asymptotic complexity is imprecise. We will revise this to emphasize constant-factor improvements under the same asymptotic complexity, supported by results in Table 2, Appendix E and F.
> 8. **Community interest.** Our contribution is to introduce large-scale self-supervised pre-training + fine-tuning for a structured conditional generation task. We believe this scalable paradigm is of broader interest beyond music. Related music-generation works have also appeared at ICML/ICLR [2–4], with recent interest further reflected by the AI4Music workshop at NeurIPS 2025.
> 9. **Impact statement and abstract clarity.** We already state that all data come from public datasets. We will make the discussion of potential effects on musicians more explicit in the final version, and clarify in the abstract that this is a conditional generation task.
>
> Overall, we hope the clarifications and additional experiment above address the reviewer’s concerns. We will make the relevant points more explicit in the final version. We thank the reviewer again for the detailed feedback.
>
> [1] BART: Denoising Sequence-to-Sequence Pre-training for Natural Language Generation, Translation, and Comprehension.
>
> [2] Graph Neural Network for Music Score Data and Modeling Expressive Piano Performance. ICML 2019, Oral.
>
> [3] Enabling Factorized Piano Music Modeling and Generation with the MAESTRO Dataset. ICLR 2019, Oral.
>
> [4] Whole-Song Hierarchical Generation of Symbolic Music Using Cascaded Diffusion Models. ICLR 2024, Spotlight.

---

> > ### Author Rebuttal · Reviewer_ab4i · 2026-04-04
> >
> > Thanks to the authors for the thoughtful response. The response did address several of my concerns. I would like to respond to two points.
> >
> > **Pre-training data contamination**. I agree that there is no _aligned_ supervision (score->performance), however, the pre-training likely contained examples of both score _and_ performance MIDI for all of the pieces in the test set. I maintain my argument that this constitutes data contamination. The references perturbation experiments do provide some evidence that the model may be generalizing, but they are not sufficient to rule out the contamination concerns entirely. E.g., the model could still be regurgitating performances with the exact timings seen during training, even if it can follow the transposition accurately.
> >
> > **Community interest**. The idea of self-supervised pre-training + fine-tuning for structured conditional generation tasks is indeed of broader interest, but is already widely known in the ICML community. The referenced ICML/ICLR music papers all included substantial methodological novelty (graph NNs, factorized acoustic/symbolic models, cascaded diffusion) to complement their domain-specific focus on music, which is not present in the current work.

---

> > > ### Author Response · Authors · 2026-04-05
> > >
> > > Thank you for your further feedback. We also appreciate that you acknowledged our previous response had addressed several concerns. Below we clarify the two remaining points.
> > > ### 1. About pre-training data contamination
> > > In addition to the difference between the pre-training and downstream tasks, and your acknowledgement that no aligned supervision signal is used during pre-training, we provide the following evidence to explain why this setting does not invalidate the evaluation.
> > >
> > > **(a) Clarification on the additional experiment.** You suggested that the model may reproduce exact timing seen during training. However, our perturbations include not only transposition but also random tempo scaling, which changes both absolute and relative timing. If the model relied on memorizing exact timing, performance should drop clearly under such perturbations. Instead, performance remains almost unchanged or is even slightly better, which supports pitch and timing generalization rather than simple memorization.
> > >
> > > **(b) Figure 6 supports that the model learns a general musical representation.** Figure 6 shows that test pieces form a continuous manifold in the latent space following historical style progression. This is more consistent with learning a structured and general musical representation than with storing individual samples.
> > >
> > > **(c) Common practice in related work.** Our setting uses denoising pre-training on unified MIDI, followed by conditional generation from score to performance. This is similar to conditional generation tasks such as machine translation. In works such as MASS [1] and BART [2], models are also pre-trained on large-scale general corpora and then fine-tuned on translation tasks, without explicitly reporting deduplication between pre-training data and test sets. Such corpora naturally contain content related to downstream tasks. However, pre-training does not provide source-to-target aligned supervision, which is the core signal of conditional generation. For this reason, such overlap is usually treated as a limitation to discuss, rather than as a reason to invalidate evaluation. Similar settings also exist in music tasks [3].
> > >
> > > Overall, these results suggest that the current evaluation primarily reflects rendering ability rather than simple memorization. We also agree to discuss this issue more explicitly as a limitation in the final version.
> > >
> > > ### 2. About community interest.
> > > Thank you for your further comments. We agree that the high-level paradigm of self-supervised pre-training + fine-tuning is already widely recognized in the ICML community. However, whether such a high-level idea is widely known does not appear to be directly related to whether a particular work lacks broader community interest. More importantly, what matters is whether the work brings non-trivial task-specific design, empirical findings, or new capabilities in a new task setting.
> > >
> > > While the related music works you mentioned adopt techniques such as graph NNs and cascaded diffusion, their contributions do not lie in inventing these components themselves, but in integrating them with music tasks to form meaningful task-specific designs. Under the same standard, our work also includes clear music-oriented novel design choices, including:
> > > - A unified score/performance MIDI representation.
> > > - An asymmetric encoder–decoder with note-level compression tailored for long-sequence modeling.
> > > - Expressive Tempo Mapping for generating editable performance outputs.
> > >
> > > Therefore, our work is not a simple reuse of an existing framework, but instead presents a coherent set of task-specific design contributions for this problem.
> > >
> > > As you noted in your original review, one **strength** you explicitly recognized was:
> > > > “Interesting architectural finding. The deep encoder / shallow decoder configuration achieving strong rendering quality with substantially improved efficiency (2.1× inference speedup, 0.6× VRAM) is a potentially broader finding worth noting, and the comparison with the symmetric 6-6 variant in Section 4.5 is appreciated.”
> > >
> > > We agree with this assessment. This point itself suggests that, in our task, introducing large-scale self-supervised pre-training can still lead to structural design insights with some degree of general relevance, rather than merely repeating a known paradigm.
> > >
> > > Therefore, the contribution of this work is not to propose a new general framework, but to demonstrate that combining large-scale self-supervised pre-training with task-specific design can bring both performance improvements and meaningful empirical insights in this new structured task. Based on this, we believe the work still has value and interest for the broader community.
> > >
> > > [1] MASS: Masked Sequence to Sequence Pre-training for Language Generation.
> > >
> > > [2] BART: Denoising Sequence-to-Sequence Pre-training for Language Generation, Translation, and Comprehension.
> > >
> > > [3] MusicBERT: Symbolic Music Understanding with Large-Scale Pre-Training.

---

### Official Review · Reviewer_QScd · 2026-03-13

**Soundness:** 3
**Presentation:** 3
**Significance:** 3
**Originality:** 3
**Overall Recommendation:** 5
**Confidence:** 4

**Summary:**

This paper presents a novel piano performance rendering model together with a training strategy and a unified tokenization that handles both score and performance MIDI. Using this tokenization, the model is first pretrained with a masked language modeling objective that predicts masked attributes from either score or performance MIDI. It is then supervisedly fine-tuned to render performance from the score. The experiments report superior performance over baselines.

**Compliance With Llm Reviewing Policy:**

Affirmed.

**Final Justification:**

I would like to thank the authors for their detailed rebuttal. Problems are addressed in the rebuttal, and I will maintain my score.

**Key Questions For Authors:**

Most questions relate to the weaknesses discussed above. It would also be helpful to include more samples from genres such as pop or jazz, if possible.

**Limitations:**

The paper should include a discussion or analysis of potential music genre bias in the performance rendering results.

**Strengths And Weaknesses:**

Strength:
1. The tokenization and the overall pipeline are very well-designed and make sense in how they help with performance rendering tasks.
2. The experiment design is clear and evaluates the model from multiple perspectives.
3. The provided listening samples are very convincing, showing massive improvement compared to existing methods.

Weakness:
1. The paper could more intuitively explain token representations and model I/O at each stage. Small, concrete examples shown on the image would improve readability.
2. From the listening samples in the provided demo link, the model appears to have limited control over global tempo. For example, sample 19.mp3, which should correspond to an Adagio tempo.

---

> ### Author Rebuttal · Authors · 2026-03-26
>
> We thank the reviewer for the high score and for the positive feedback on our method design, experimental quality, and demo results. Below we respond to the reviewer’s suggestions and concerns:
>
> 1. **Clarity of representation and stage-wise inputs/outputs.** We appreciate this helpful suggestion. The paper already includes concrete examples of the unified representation in Appendix A.2.2 and Figure 8. To further improve clarity, we will add a token legend to Figure 2 in the camera-ready version, so that readers can more easily interpret the inputs and outputs at each stage.
>
> 2. **Global tempo control.** We appreciate this careful observation. At present, the model mainly uses the actual tempo of the input score MIDI as the reference for rendering global tempo. For the sample 19.mp3 you mentioned, we checked that the input score MIDI itself is relatively fast, with a duration of only 1 minute 20 seconds, although this may of course disagree with the original composer's tempo marking, so the model renders it at a faster tempo. We acknowledge that the current framework does not provide an explicit interface for global tempo control. The Expressive Tempo Mapping in the paper converts the generated result into an editable tempo curve, so globally scaling the entire tempo curve in a DAW may help alleviate this issue. We will state this more clearly as a limitation and treat explicit tempo control as future work.
>
> 3. **Genre coverage and style bias.** We agree that adding pop or jazz examples would strengthen the paper's claim about style generalization. If the paper is accepted, we will add pop or jazz demo examples to the open-source repository. We also agree that discussing potential style bias can help readers better understand the current boundaries of the model’s capabilities, and we will include a dedicated discussion on this aspect in the appendix of the camera-ready version.
>
> We again thank the reviewer for the supportive comments and careful review.

---

> > ### Author Rebuttal · Reviewer_QScd · 2026-04-04
> >
> > Thank you for the rebuttal.
> >
> > Regarding point 1: It is indeed necessary to include examples in Figure 2, not just in the appendix. Figure 2, in its current form, is not fully intuitive: the meaning of the color arrangement is not immediately clear. Incorporating examples into the main figure would help convey the core idea more concretely and significantly improve clarity.
> >
> > Regarding point 2: The explanation is reasonable and does not affect the overall significance of the method. However, the current presentation may still cause confusion for readers.
> >
> > Regarding point 3: Please incorporate the revision in the final manuscript.
> >
> > I have no further questions and will maintain my current score.

---

> > > ### Author Response · Authors · 2026-04-04
> > >
> > > Thank you for your constructive comments, which have helped us further improve the quality of the paper. We sincerely appreciate your support again.

---

### Decision · Program_Chairs · 2026-04-30

**Decision:**

Accept (regular)

**Comment:**

The paper proposes a method for self-supervised learning of expressive piano performance based on >100k of MIDI data, as well as a Transformer-based encoder-decoder model for generating such performances, and a rendering model that the authors claim as state-of-the-art.

Reviewers QScd and b9Tf considered the paper technically solid, and see as strenghts the data pipeline and experiment design (including ablations), as well as the provided samples output by the model. The weaknesses pointed out by QScd are minor, mostly on clarification and a limitation on the global tempo control by the model. The authors addressed QScd comments and seem to have incorporated the suggestions in the final version. The weakness pointed out by b9Tf is the small number of listeners in the subjective evaluation, and limited amount of data per listeners, which the authors agreed is a limitation of their study and have acknowledged should be seen only as preliminary evaluation.

Reviewer ab4i has a very contradicting review compared to the other two reviewers, while claiming a high level of confidence. The concern with data contamination during pre-training is valid, but the reviewer also raises concerns about the paper that seem to have been addressed by the authors in the original version (with some other concerns addressed in the final version). The reviewer also claims the paper is of limited relevance to the broader ICML community, to which I both agree and disagree with. While there is not significant novelty in the unsupervised pre-training methodology, the task specific design is of interest to a growing audience in ML research interested in AI applications in music.

In order to take into account the concerns of reviewer ab4i but still value the contribution of the authors and how the other two reviewers rated this paper, I suggest a weak accept score.